# Short-Term Impact of AC Harmonics on Aging of NiMH Batteries for Grid Storage Applications [note 1]

**DOI:** 10.3390/ma14051248

**Published:** 2021-03-06

**Authors:** Jenny Börjesson Axén, Rudi Soares, Oskar Wallmark, Peter Thelin, Erika Widenkvist Zetterström, Göran Lindbergh

**Affiliations:** 1Department of Chemical Engineering, Division of Applied Electrochemistry, KTH Royal Institute of Technology, SE-10044 Stockholm, Sweden; gnli@kth.se; 2Nilar AB, Bönavägen 55, SE-80647 Gävle, Sweden; peter.thelin@nilar.com (P.T.); erika.widenkvistzetterstrom@nilar.com (E.W.Z.); 3Electrical Machines and Drives Laboratory, Department of Electric Power and Energy Systems, KTH Royal Institute of Technology, SE-10044 Stockholm, Sweden; rhsoares@kth.se

**Keywords:** accelerated aging, batteries, battery aging, energy storage systems, NiMH batteries, power conversion harmonics, power system harmonics

## Abstract

Batteries in energy storage systems are exposed to electrical noise, such as alternating current (AC) harmonics. While there have been many studies investigating whether Lithium-ion batteries are affected by AC harmonics, such studies on Nickel Metal Hydride (NiMH) batteries are scarce. In this study a 10 Ah, 12 V NiMH battery was tested with three different harmonic current frequency overlays during a single charge/discharge cycle: 50 Hz, 100 Hz, and 1000 Hz. No effect on battery internal temperature or gas pressure was found, indicating that NiMH battery aging is not affected by the tested harmonic AC frequencies. This can reduce the cost of energy storage systems, as no extra filters are needed to safeguard the batteries. Instead, the capacitive properties of the batteries give the possibility to use the battery bank itself as a high pass filter, further reducing system complexity and cost.

## 1. Introduction

Industrial scale Nickel Metal Hydride (NiMH) batteries have many different uses, both as stationary systems and in traction applications. Stationary applications include, e.g., uninterruptible power supply (UPS), smart grid energy storage, fast charging buffer for electric vehicles, and home storage. In many of these applications, the batteries are continuously exposed to direct current (DC) loads that contains a certain amount of harmonics. This can be caused by the converter itself, e.g., by the control method and the switching frequency of the converter, and/or through incomplete filtering of harmonics present in the grid voltage [1,2]. In some systems, another source of superimposed alternating currents (AC) can be the pulsating AC power of a single-phase electric motor and other single-phase AC loads, which are connected to the battery via an unfiltered DC/AC-inverter [3].

Previous work on Li-ion batteries for traction systems disagree on what effect AC frequency overlay has on the batteries. Some studies show no effect on aging when cycling with an AC frequency overlay [4,5,6,7]. The studies that do show an effect disagree on what frequencies are causing it. Uddin et al. find that the capacity fade and the impedance increases with increasing frequency [8], while other studies point to the lower frequencies as the culprit in changing the battery behavior [5,9,10]. In contrast to these studies that place importance on the frequency of the overlaid harmonics, Juang et al. conclude that it is the amplitude that is the main contributing factor, with higher amplitudes having a higher impact irrespective of frequency [11]. Osswald et al. investigate the current path taken through the cell using impedance measurements and conclude that it varies with overlaid frequency, identifying the risk of uneven aging caused by AC overlay [12]. However, studies on how batteries of the NiMH chemistry are affected are scarce, with one found that studies impact on capacity [13]. Understanding the impact of system related AC frequency overlay on NiMH batteries is important to ensure the longevity of affected battery energy storage. The study presented in this paper aims to investigate the short-term effect of DC current overlaid with AC frequencies on the behavior of a NiMH battery, to assess whether battery life is affected.

In a NiMH battery, the temperature and gas pressure behaviors are closely related to the aging mechanisms, both as causes and symptoms. This allows for investigation of aging effects of AC frequency overlay through studying the temperature and gas pressure behaviors during individual cycles, which simplifies testing and decreases testing time. Studies investigating only a single cycle [9] or a part of one [11] have been made for different Li-ion battery types. To apply similar methods to a NiMH battery, the behavior specific for this battery type needs to be considered. The primary aging mechanism in the NiMH battery is corrosion of the negative electrode, which leads to increased resistance. The process is accelerated by high temperature as well as a high oxygen partial pressure in the cell. Increases in temperature and/or gas pressure can therefore be indicators of the battery being harmfully affected by the frequency overlay [14]. Taking these mechanisms into account, this study uses the temperature and gas pressure behavior of a NiMH battery when subjected to AC frequency overlay to assess the effect on battery aging.

## 2. Materials and Methods

In this study, the battery tested was a starved electrolyte configuration Nilar 10 Ah EC 10-cell battery (Gävle, Sweden) with a nominal voltage of 12 V. A pressure sensor (P51-100-A, ±0.5% inaccuracy, SSI Technologies, Inc., Janesville, WI, USA) measured the inner gas pressure of the battery through a gas channel which is shared by all cells. The temperature sensor (PT100, ±0.4 °C inaccuracy, IST, Wattvil, Switzerland) was placed in an electrically insulated slot in the contact plate on the negative side of the battery case. All tests were carried out at room temperature, 22 ± 1 °C.

The experimental setup contains a voltage source converter (VSC), allowing the superposition of modulated sinusoidal current waveforms with frequencies up to 2 kHz on top of a DC current. A cDAQ data acquisition system (cDAQ 9174, National Instruments, Austin, TX, USA) exchanges data with a computer.

A layout of the experimental setup is shown in Figure 1. A more detailed explanation can be found in Appendix A, with Figure A1 providing a picture of the experimental setup, and the setup is described in further detail by Soares et al. [15].

To study the effect of frequency overlay on the NiMH battery, a battery at beginning of life was cycled one charge/discharge-cycle with constant DC current for each frequency. The current is given as a C-rate, where C is the current needed to discharge the battery fully in 1 h. The tested cycles consist of a charge immediately followed by a discharge and is defined by an upper and lower voltage boundary, where the lower boundary is 1.000 V/cell in all cases. In this study two different C-rates were tested, 0.3C, with an upper voltage boundary of 1.475 V/cell, and 0.5C, with an upper voltage boundary of 1.505 V/cell. For each C-rate there were three different frequencies tested as well as a reference test without any frequency overlay. The frequency overlay was sinusoidal with an amplitude of 70% of the tested DC C-rate. The three frequencies selected were 50 Hz, 100 Hz, and 1000 Hz. The first two were chosen as they are possible frequencies occurring on the DC side when a battery system is connected to the grid through an AC/DC-converter [1,2]. To study the effect of a higher frequency current oscillation on battery behavior, 1000 Hz was chosen to make the results comparable to previous studies [4].

The tested battery module was characterized after the frequency tests, using Electrochemical Impedance Spectroscopy (EIS, Zahner, Kansas City, MO, USA). The EIS measurement was done at room temperature in a galvanostatic mode with a 100-mA amplitude and a range of 10 mHz–50 kHz.

## 3. Results

To evaluate the effect of frequency overlay on the NiMH battery, its temperature, voltage, and gas pressure behaviors were studied. These behaviors are presented in Figure 2. Observing the temperature behavior of the battery at the different test conditions shows no significant effect of adding an AC overlay to the DC current during charge. 

Similarly, the temperature behavior during discharge is unaffected. Likewise, the voltage characteristics do not change as the battery is subjected to an AC overlay, independent of the frequency of the overlay, neither for charge nor discharge. Finally, the gas pressure behavior of the battery shows no effect of introducing an AC current overlay. Each DC current rate has a gas pressure behavior that is replicated for all tested frequencies. The slight differences seen between the tests cannot be significantly tied to any of the specific AC frequencies and are likely a result of natural fluctuations in test conditions.

## 4. Discussion

In this study, the effect of AC current overlay on the NiMH battery is examined at the beginning of life (BOL) by observing voltage, temperature and gas pressure behaviors during a single charge/discharge-cycle, as they can give a good indication on whether aging will be affected. It is known that exposure to singular events can negatively affect the life span of batteries, and short-term changes can give an indication whether this has occurred. As none of the measured properties in the NiMH battery were affected by the frequencies used, it is unlikely that the tested AC-frequency overlay increases the rate of aging of the battery.

The major aging mechanism for a NiMH battery is corrosion of the negative electrode which consumes metal, creating metal hydroxide [14]. A metal hydride stores hydrogen in free spaces between the metal atoms in a crystal lattice, so called intercalation sites. The oxidation of the metal to metal hydroxide destroys some hydrogen storage sites, and the hydrogen produced in the oxidation reaction also occupies additional intercalation sites. Both mechanisms reduce the total amount of available hydrogen storage sites. For each corroded site, a bit more than twice the hydrogen storage capacity is lost [16]. In addition, the corrosion process consumes water from the electrolyte causing dry-out in the separator, which is a porous polymer layer separating the electrodes, increasing the electrical resistance of the battery [14]. The corrosion rate of the negative electrode is increased by higher temperatures [14,16,17,18,19].

A NiMH battery is designed to be capacity limited by the positive electrode with a negative electrode over-capacity. As the negative electrode material is oxidized, the capacity balance of the electrodes is shifted, eventually causing the battery to instead be limited by its negative capacity. With the negative electrode unable to receive all of the hydrogen produced during the charge process, the gas pressure in the battery during charging is increased. This causes the gas pressure limit to be engaged earlier in the charge cycle, decreasing the usable capacity of the battery. A slower pressure drop during the discharge part of the cycle can indicate a presence of gaseous hydrogen, since the oxygen recombination rate is lower in an oxygen hydrogen mix than in an environment free of gaseous hydrogen [20].

Just like temperature, the gas pressure balance can affect the aging of the battery. An increase of oxygen partial pressure aggravates the corrosion of the negative electrode material. A discharged negative electrode is more vulnerable to this process, since the charged electrode is protected by the presence of intercalated hydrogen. Changes in pressure behavior with increased pressure levels during charge can indicate problems for the negative electrode to intercalate hydrogen, increased oxygen production during charging, as well as reduced gas recombination rates. As oxygen pressure drives the corrosion reaction and hydrogen intercalation limits the corrosion reaction, all three scenarios as described above could lead to a higher corrosion rate of the negative electrode material [21].

Considering the aging behavior of the NiMH battery type as discussed in previous paragraphs, the lack of effect on temperature or gas pressure found in this study indicates that the aging is not aggravated by the superimposed AC-frequencies tested.

While this study shows no effect of AC frequency ripple on a NiMH battery at beginning of life, there are further tests to be made to ensure that AC frequency overlay is not a factor that affects the long-term performance of the battery. To confirm that the aging is unaffected, long-term cycling experiments could be carried out. In addition to temperature and gas pressure, discharge capacity and resistance should also be studied to evaluate the effect on aging.

When describing the electrical properties of an electrochemical system it is useful to refer to EIS measurements. EIS is used to characterize batteries regarding electrical resistance, electrode surface area, reaction kinetics, and diffusion properties. In this study only one EIS measurement is performed, at 50% SOC, but performing EIS measurements over the whole SOC range is a good way to map battery characteristics dependence on SOC [22]. Looking at a Nyquist plot of the tested battery, Figure 3a, it consists of an inductive region above 7 kHz, a high frequency region between 7 kHz and 0.5 Hz, and a low frequency region Warburg diffusion tail below 0.5 Hz. All of the AC-overlay frequencies tested in the present study are located in the high frequency region. Previous studies made on Li-ion use EIS to analyze AC frequency overlay response. Uno et al. argue that low frequencies, slow enough to activate the charge transfer reactions, are the cause of aging due to side reaction activity [10]. Another study expanding on this theory claims that the critical frequency, dubbed the corner frequency, below which this process happens, can be found at the top of the arc of the high frequency region [5]. Considering that the frequencies used in this study are all higher than the corner frequency, the fact that no effect from the AC frequencies can be seen on the battery is in accordance with both of these theories. A common analysis method of EIS data is to adapt an electrical circuit chosen to approximately model the internal processes of the system, i.e., an equivalent electrical circuit. For battery systems the chosen circuit is commonly a variation of the Randles circuit [23], where the chemical reactions and mass transport properties of the battery are represented by a capacitance connected in parallel to a resistance and a Warburg element—a representation of the electrochemical mass transport—in series. This parallel circuit is then combined with a resistance in series representing the ohmic resistance, Figure 3b. Considering the capacitive properties of the battery illustrated by the Randles circuit, the battery should behave as a high pass filter. Since the current ripple investigated in this study has been found to have no negative effects on the battery, a NiMH battery bank could be used as a high pass filter in a system where there is a risk of superimposed frequencies. This would decrease system cost and complexity for systems where frequency overlay is common. To further investigate this and confirm that the battery would behave as a high pass filter over the whole SOC range, more EIS measurements could be made at different SOC levels.

## 5. Conclusions

When subjecting a NiMH battery to three different AC-overlay frequencies, 50 Hz, 100 Hz and 1000 Hz, no short-term effects could be seen on the battery voltage, temperature, and internal gas pressure behavior. These results show that there are no signs that the battery is aged by the frequencies tested. This as the primary aging mechanism is accelerated by high oxygen pressure and elevated temperature. Due to this result, no filters would be needed to safeguard the batteries in a grid connected energy storage system. In addition, the capacitive properties of the battery make it behave like a filter, and so a battery bank could be used in lieu of added filters in an energy storage system, further decreasing system cost and complexity.

## Figures and Tables

**Figure 1 materials-14-01248-f001:**
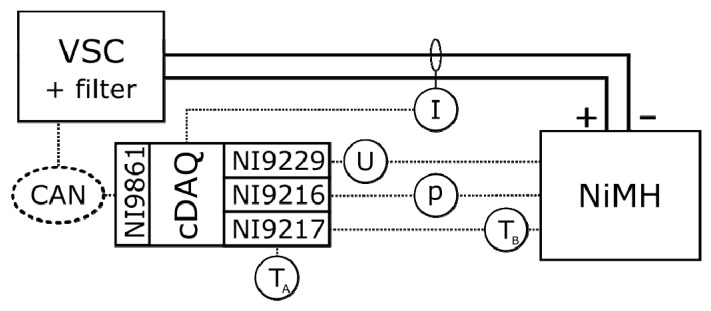
Schematic representation of the experimental setup. Numbers in cDAQ corresponds to used acquisition cards.

**Figure 2 materials-14-01248-f002:**
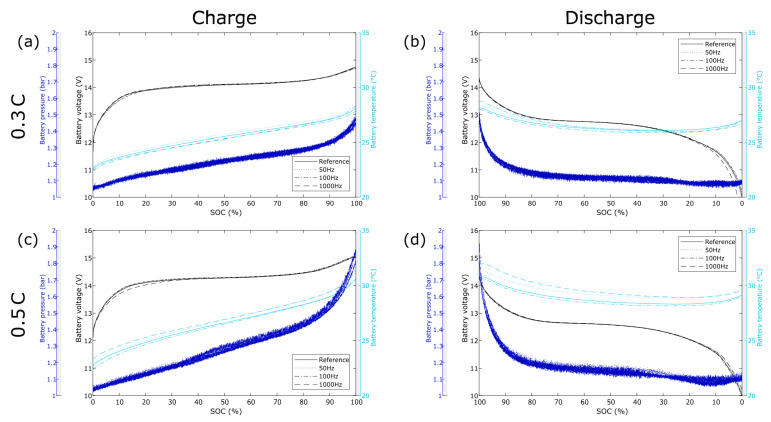
NiMH battery voltage, temperature, and absolute gas pressure vs. State of Charge (SOC) for different overlay frequencies. The sinusoidal AC-amplitude is 70% of the C-rate. Ambient temperature: 22 ± 1 °C. (**a**) Charge 0.3 C; (**b**) Discharge 0.3 C; (**c**) Charge 0.5 C; (**d**) Discharge 0.5 C.

**Figure 3 materials-14-01248-f003:**
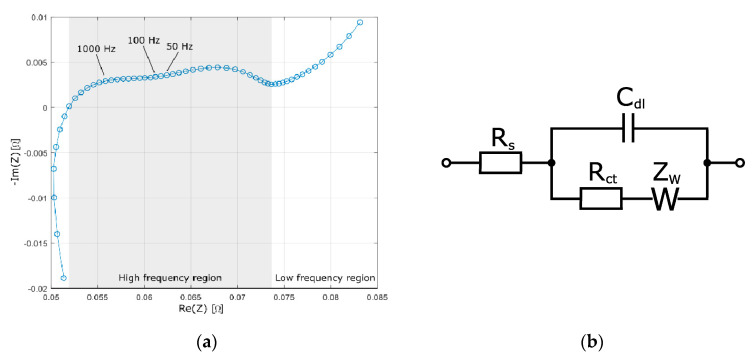
(**a**) Nyquist plot of tested battery, galvanostatic measurement with 100 mA amplitude, and range 10 mHz–50 kHz. Measured after tests, at room temperature, 22 ± 1 °C. (**b**) A Randles circuit, often used to simulate battery behavior. Where R_s_ is the ohmic resistance, C_dl_ is the electrochemical double layer capacitance, R_ct_ is the charge transfer resistance, and Z_W_ is the Warburg element.

## Data Availability

Restrictions apply to the availability of these data. Data are available on request from the corresponding author with the permission of Nilar AB.

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
