# Peer review of "Short-Term Impact of AC Harmonics on Aging of NiMH Batteries for Grid Storage Applications"

_materials, 2021, doi:10.3390/ma14051248_

Round 1

Reviewer 1 Report

Problem of AC harmonics on aging of NiMH batteries especially for grid storage applications is actual and very important. However, paper is too short and many important facts are missing at current version and must be added in my opinion.
Introduction is very short, and looks more like conference paper introduction. It is good to extend it little bit. Some newer works on the topic could be also added e.g. temperature profiling of Li-ion batteries,
doi: 10.1109/SIELA49118.2020.9167036
and battery deformation measurements,
doi: 10.1109/SIELA49118.2020.9167109
In Materials and Methods, must be described NI devices measurement accuracy, it is recommendable to add the visual outlook of measurement setup. Probably harmonic spectrums of AC currents could be presented somehow.
Figure 1. has an enormous font, please reduce the text size in the blocks.
Figure 2 Blue text is unreadable for me.
Temperature and pressure results measured on outer battery enclosure are not very useful.
Finally, EIT results are missing, add fitted circuit element values of Figure 3-b. Consider impedance variation with frequency. Consider filtering capability over the battery capacity, etc.
Conclusion is somehow doomed, and need to be rewritten in much positive manner, pointing research strong points and outcomes.

Author Response

Dear reviewer,

Thank you for your valuable time and your insightful comments. We’ve tried to answer each of your concerns below, in the order that you brought them up.

  • You comment that the introduction is short. Since the paper is a communication, we felt the need to not overburden the introduction but instead focus on the vital information. To write a short introduction was an active decision to keep the balance in the paper, especially since the result section is short. We wanted a paper that was short and to the point, something that the other reviewers regarded favorably, and we fear that adding more to the introduction would take away rather than add value to the paper. Regarding the papers you suggested, while interesting and useful for Li-ion batteries, those techniques would be difficult to use for NiMH batteries. The deformation of the battery over the charge cycle is not something that happens for the bipolar NiMH battery type, as the expansion of one electrode is matched by a shrinkage of the other, and what little difference there is, is buffered by the porous separator. Regarding the external temperature measurement technique, it is very interesting, but would unfortunately be impossible to use for the batteries used in this study. The Nilar EC is designed to be modularly put together into packs, and consequently the top and bottom of the battery is covered by a substantial end piece which would diffuse the results from an IR camera. The temperature sensor that we use in this study is placed closer to the cell and will therefore be more accurate. We’ve attached a link to the product specifications: https://www.nilar.com/products/battery-pack/ A picture of the battery used for the testing can also be find in the added supplementary information which brings us to:
  • For the same reason that the introduction was kept short, we did the same to the material and methods section. We would preferably keep it that way, but the information you requested has been added to the Appendix. Hopefully this should make it a bit clearer and be to your satisfaction.
  • The font-size in Figure 1 has been reduced. In addition, we discovered that the lines in the figure did not connect the acquisition cards to the right sensors. This has been corrected.
  • The saturation of the blue text in Figure 2 has been lowered, hopefully this should have increased the legibility.
  • Here we think that there has been a misunderstanding. The pressure is measured through a feature on the battery called the “common gas channel” which is connected to the gas space of each cell. Hence the pressure measured is the internal gas pressure of the battery, rather than the external pressure usually measured on Li-Ion. Considering that oxygen evolution is a distinct feature of the NiMH battery chemistry, measuring the internal battery pressure is useful for many things, especially as a safety feature to terminate charging before any runaway in pressure or temperature can occur. Regarding the temperature, it is measured as close to the cell compartment as the manufacturer is able. The channel for the temperature sensor is located on the contact plate of the bipolar battery module. This leaves 1-2mm of plastic between the senor and the 10 cells inside the module. This is the same location that is used for the temperature sensor in the battery systems, with the temperature signal used both in the end of charge algorithm as well as a safety feature. We have attempted to clarify what pressure and temperature that are measured in the text.
  • Regarding the EIS fitting, we have chosen not to adapt an EEC model to the data, since this would draw focus from main story of the article. In addition, the authors are currently working on producing a physics-based battery model and have chosen to focus on the modelling and characterization aspects of the battery there. The reason to include the EEC in the text was to give a better understanding of the electrical battery behaviors to professionals working with electrical power systems, and to strengthen the argument that the battery bank in the energy storage system can be used as a capacitive filter. The impedance variation with frequency should be visible in the Nyquist plot, but it might be unclear. There is also a section discussing the vulnerability to different frequencies with base in the Nyquist plot and findings from literature already present in the discussion. Considering filtering capability over the battery capacity, this would require EIS measurements over the whole SOC span, while these measurements were only made at 50% SOC. It would be a good topic for a future study, but is not further investigated here to keep the shorter format of the paper. We have added a sentence at the end of the discussion section to this effect.
  • In order for the conclusion to feel less gloomy, we have rewritten it in a more positive tone.

We hope that these changes will make the paper more to your liking, and that the explanations are to your satisfaction.

Sincere Regards,

Jenny Börjesson Axén – Corresponding author

Reviewer 2 Report

The paper "Effects of AC harmonics on aging of NiMH batteries for grid storage applications" is very clearly written and provides a clear layout of the performed experiments together with a strong justification of the obtained results. A well formulated outlook on possible future investigations, which should focus on long-term effects of AC harmonics on ageing, conclude this paper.

I would like to propose only two very minor modifications:

1) Fig. 3 contains indices of the equivalent circuit parameters that are undefined. Please add their meaning to the figure legend or figure caption.

2) I am wondering whether effects of aging that influence the characteristic frequencies indicated in Fig. 3(a) could be described efficiently in terms of an analytic fractional-order system model. They are well-known from the world of Li-Ion batteries. Maybe adding some related references in the frame of the outlook could be reasonable. However, I'll leave this up to the authors:

C. Zou, et al.: A Review of Fractional-Order Techniques Applied to Lithium-Ion Batteries, Lead-Acid Batteries, and Supercapacitors Journal of Power Sources, 390, 286-296, 2018.

D. Andre, et al.: Characterization of High-Power Lithium-Ion Batteries by Electrochemical Impedance Spectroscopy: I. Experimental Investigation, Journal of Power Sources,
196(12), 5334-5341, 2011.

Rauh, Andreas; Kersten, Julia: Verification and Reachability Analysis of Fractional-Order Differential Equations Using Interval Analysis, Proceedings 6th International Workshop on Symbolic-Numeric Methods for Reasoning about CPS and IoT (SNR 2020), Electronic Proceedings in Theoretical Computer Science 331, pp. 18-32. 2021. DOI: 10.4204/EPTCS.331.2

Author Response

Dear reviewer,

Thank you for your valuable time and your insightful comments. We are very happy that you found the paper to your liking. We’ve tried to address your two concerns below, in the order that you brought them up.

1) We have added the definitions of the equivalent circuit parameters to the figure caption.

2) Concerning fractional-order system models it is an interesting approach, and the EEC that is presented in the paper does contain a Warburg element, which is considered to be a fractional-order element. Modelling the mass-transport properties of the battery in an EEC is neigh impossible without using some kind of fractional-order element, and the Warburg element is derived from the mass transport equations found in electrochemical theory. But it is difficult to see as the convention used was not the same as the review article by Zou that you recommended (and that the parameters are not defined in the figure caption as you pointed out earlier). In fact, if you look closely at the 3 b) diagram in the Zou article, it is the same circuit that has been pictured in this paper. Considering the application of battery models, the purpose of the equivalent circuit featured in this paper was never to characterize the battery, but rather to provide understanding to the readership that come from the system side and are used to work with electrical components. Three of the authors work with creating battery energy storage systems and have noticed how difficult it is to communicate from electrochemistry to electrical engineering (And also the other way around!). So by including an EEC-diagram we hope to provide substance to the claim that the battery has capacitive properties and can be used as a filter in an battery energy storage system to shield the other components.

We have chosen to include the Andre article to strengthen the discussion about EIS. As the corresponding authors main research is modelling of NiMH batteries and we are about to publish an article on a physics-based model for voltage estimation, we want to keep the two topics a bit separated and focus on the practical implications of AC-frequency overlay on ageing for this paper. So please stay tuned if you find modelling and characteristics of NiMH batteries interesting!

We would like to comment the added part in the appendix; one of the other reviewers wished for us to include more details on the experimental set up, and so we put that in the appendix to still keep the concise nature of the main text.

We hope that the changes that we have made has improved the paper from your perspective and look forward to your further comments.

Sincere Regards,

Jenny Börjesson Axén – Corresponding Author

Reviewer 3 Report

This short communication presents a convincing preliminary experiment to assess the impact of AC ripple overlay (@ 50, 100 and 1000 Hz) on the ageing of an industrial NiMH battery bank consisting of 10 cells. It is made clear in the article that this is a short-term experiment which only conducts one charge-discharge cycle to investigate aging. Nevertheless, the fact that no indications of corrosion are found is an interesting result, and one which warrants further investigation with longer test duration, as the authors suggest. The paper is well written, concise yet informative, and the reviewer enjoyed reading it. For this reviewer, the short communication could be published almost as-is.

The only suggestion I would make to authors would be to incorporate 'a preliminary study', 'short-term impact' or 'single-cycle' in the title somewhere, to better frame the work and keep the scope to a preliminary results. This may also make it easier when authors are ready to publish a follow-up study with deeper results.

Author Response

Dear reviewer,

Thank you for your valuable time and your insightful comments. We are very happy that you found the paper to your liking.

We are very grateful for your suggestion that we change the title to better frame that the work was done as a short-term/preliminary study. We have elected to use “short-term impact”, as we feel that that formulation suits the work best.

We would like to comment the added part in the appendix; one of the other reviewers wished for us to include more details on the experimental set up, and so we put that in the appendix to still keep the concise nature of the main text.

We hope that the changes that we have made has improved the paper from your perspective and look forward to your further comments.

Sincere Regards,

Jenny Börjesson Axén – Corresponding Author

Round 2

Reviewer 1 Report

I accept most of the authors claims in the response letter and I see many positive changes.

In Title “Short-term impact of AC harmonics on aging of NiMH batteries for grid storage applications”, Short-term semantically conflicts with aging.

I still believe that Conclusion can be improved by adding more context of usage, limitations and future work perspectives.

Author Response

Dear reviewer,

We are happy that you see improvements from the last version. Below we will go through your two suggestions.

While short-term semantically conflicts with aging, the way this title was intended to be understood was that the AC harmonic exposure is short-term, and that the impact of this exposure on aging is investigated. The reason that we added “short-term” to the title was that it was suggested by one of the other reviewers as a way to clarify the content in the paper and open up for future continued long-term studies. We have added a sentence in the discussion to clarify the connection between short-term exposure and long-term adverse effects. This should hopefully give a better framing of the problem.

As for the second comment that the conclusion is still a bit meager, we’ve tried to expand it by clarifying why a battery bank can be used as a filter in an energy storage system. However, we do not feel that suggestions for further studies belong in a conclusion. The reader will read seek the conclusion out as one of the first things they reads in order to find out if the paper contains what they are looking for, and so we want to keep it concise and focused on the conclusions we can draw from the results. Instead we have left the discussion on further studies in the discussion section.

Best Regards,

Jenny Börjesson Axén – Corresponding Author